# Development of Novel Real-Time Radiation Systems Using 4-Channel Sensors

**DOI:** 10.3390/s20092741

**Published:** 2020-05-11

**Authors:** Yohei Inaba, Masaaki Nakamura, Masayuki Zuguchi, Koichi Chida

**Affiliations:** 1Course of Radiological Technology, Health Sciences, Tohoku University Graduate School of Medicine, 2-1 Seiryo, Aoba, Sendai, Miyagi 980-8575, Japan; nakamura.masaaki@nifty.com (M.N.); qqrm6wq9k@arrow.ocn.ne.jp (M.Z.); chida@med.tohoku.ac.jp (K.C.); 2Department of Radiation Disaster Medicine, International Research Institute of Disaster Science, Tohoku University, 468-1 Aramaki Aza-Aoba, Aoba, Sendai, Miyagi 980-0845, Japan

**Keywords:** developed dosimeter system, disaster medicine, fluoroscopically guided intervention, multi-channel sensor, medical radiation dose, radiation skin injuries, real-time radiation sensor

## Abstract

Radiation-related tissue injuries after medical radiation procedures, such as fluoroscopically guided intervention (FGI), have been reported in patients. Real-time monitoring of medical radiation exposure administered to patients during FGI is important to avoid such tissue injuries. In our previous study, we reported a novel (prototype) real-time radiation system for FGI. However, the prototype sensor indicated low sensitivity to radiation exposure from the side and back, although it had high-quality fundamental characteristics. Therefore, we developed a novel 4-channel sensor with modified shape and size than the previous sensor, and evaluated the basic performance (i.e., measured the energy, dose linearity, dose rate, and angular dependence) of the novel and previous sensors. Both sensors of our real-time dosimeter system demonstrated the low energy dependence, excellent dose linearity (R^2^ = 1.0000), and good dose rate dependence (i.e., within 5% statistical difference). Besides, the sensitivity of 0° ± 180° in the horizontal and vertical directions was almost 100% sensitivity for the new sensor, which significantly improved the angular dependence. Moreover, the novel dosimeter exerted less influence on X-ray images (fluoroscopy) than other sensors because of modifying a small shape and size. Therefore, the developed dosimeter system is expected to be useful for measuring the exposure of patients to radiation doses during FGI procedures.

## 1. Introduction

Radiation-related tissue injuries after medical radiation procedures, such as fluoroscopically guided intervention (FGI), have been reported in patients [1,2,3,4,5,6,7,8,9,10,11,12,13,14]. In addition, these procedures are often performed repeatedly [4]. Thus, real-time monitoring of medical radiation exposure administered to patients during FGI is important to avoid such tissue injuries. [15,16,17,18,19,20]. 

The International Commission on Radiological Protection (ICRP) provided recommendations concerning optimizing patient doses using diagnostic reference levels (DRLs) because of increasing concerns about skin radiation dose levels in FGI [2]. Furthermore, the ICRP has provided importance to monitor, in real-time, whether the threshold doses for tissue reactions are being approached or exceeded for radiation protection during FGI [3]. To protect radiation-related tissue injuries, the maximum radiation tissue dose (MTD) measurement in real-time is essential.

Currently, the passive dosimeters (including thermoluminescent dosimeters (TLDs), radiophotoluminescence glass dosimeters (RPLDs), and optically stimulated luminescence dosimeters (OSLDs)) do not measure and monitor radiation doses in real-time [21,22]. On the other hand, although there are skin dose monitors (SDM; McMahon Medical, Los Angeles, California, USA) and patient skin dosimeters (PSD; Unifors Co., Ltd., Billadal, Sweden) as real-time dosimeters, the SDM (only a single channel sensor) sensor has been a toxic substance and sales stop, and the PSD sensor (3-channel sensors), using a semiconductor, has been markedly visible on fluoroscopic images [22,23,24]. Thus, no feasible real-time dosimeter with a multi-channel sensor is available for FGI procedures. Therefore, we developed a novel radiation dosimeter with 4-channel real-time sensors. In this study, we evaluated the usefulness of the developed real-time radiation dosimeter systems.

## 2. Materials and Methods

### 2.1. 4-Channel Real-Time Dosimeter System

Previously, we reported a photoluminescence real-time radiation dosimeter system using a red emission phosphor (Y_2_O_2_S: EU, SM) with 4-channel nontoxic sensors for real-time monitoring of patient MTD during FGI procedures [23,24,25,26]. Our developed, previous radiation sensor has demonstrated low sensitivity to radiation from the side and back, although it had a marginal influence on X-ray images and high-quality fundamental characteristics [25,26]. Thereby, in this study, we have significantly improved the previous radiation sensor regarding the angular dependence by modifying the shape of the phosphor from a disk to a circular cone (Figure 1 and Figure 2). Our novel real-time radiation system consists of a maximum of 4-channel nontoxic phosphor sensors, an optical fiber cable for 2.5 m, a photodiode, and a digital display that indicates the radiation dose value, such as cumulative absorbed dose (mGy) and absorbed dose rate (mGy/min) (Figure 3 and Figure 4). The maximum 4-channel sensors can measure the MTD by sticking to a patient’s skin undergoing FGI, whereas the SDM has only a single sensor. Moreover, the novel dosimeter does not need to process the annealing (pre-processing) or readout operation (post-processing), such as the TLDs, RPLDs, and OSLDs. Consequently, we expect the widespread use of our novel dosimeter system for patient MTD measurement during FGI.

### 2.2. Dosimeter Response Characteristics

The present study evaluated the basic characteristics of the newly developed 4-channel sensors, such as uniformity and reproducibility among four sensors, energy-, dose-, and dose rate-dependence. An X-ray system for irradiation was used, the DHF-155H (Hitachi, Tokyo, Japan). To compare the radiation dose measurement of our novel sensors, we used our previous sensor and the SDM. A 6-mL thimble ion chamber (model 9015, Radcal Corporation, Monrovia, California, USA) was used as a reference dosimeter, and was calibrated at Japan Quality Assurance Organization (JQA; Tokyo, Japan) in a secondary standard irradiation field. The geometric arrangement of our manner was a distance of 100 cm from the source of the X-ray tube to the measurement dosimeter in free air. The irradiation field was 20 cm in diameter on the surface of the dosimeters.

The uniformity of the novel four sensors was carried out without the calibrated factor at fixed positions and at the same time for each measurement; these placed at approximately 2 cm intervals. The reproducibility was sequentially executed by relocating each of the four sensors. The X-ray conditions of these experiments used 80 kV tube voltage, 250 mA tube current, and 400 ms irradiation time. These evaluated values were obtained from the coefficient of variation (CV) by ten measurements.

The energy dependence experiment of the dosimeters was carried out using 60, 70, 80, 90, 100, 110, 120, and 130 kV (respectively, 29.1, 30.6, 32.5, 34.3, 36.0, 37.8, 39.5, and 41.0 keV effective energy) tube voltage, 250 mA tube current, and 400 ms irradiation time for the same exposure time. The evaluated values were obtained by averaging three measurements for each dosimeter. The ion chamber results were used as a reference and all the values were normalized to its 90 kV dose.

The dose dependence of the dosimeters was carried out using 80 kV tube voltage, 250 mA tube current, and 400 ms irradiation time, in which the range of radiation exposure was from 0.2 to 470 mGy. This evaluated value was obtained from one measurement for each dosimeter in comparison with the value of the ion chamber.

The dose rate dependence of the dosimeters was carried out at 13 different measuring points, with tube current ranging from 10 to 500 mA, under 80 kV tube voltage, and 400 ms irradiation time, in which the range of radiation exposure rate was from 0.3 to 50 mGy/s. The evaluated values were obtained by averaging three measurements for each dosimeter. The ion chamber results were used as a reference and all the values were normalized to its 6.5 mGy/s dose.

### 2.3. Angular Dependence and Fluoroscopic Image

The X-ray unit used for angular dependent measurement and the fluoroscopic image was the Infinix Celeve-I (Toshiba Medical Systems Corporation, Otawara, Japan). 

The angular dependence of the dosimeters was carried out, measuring from 0° to ± 180° in the horizontal and vertical axis, using 80 kV tube voltage, 50 mA tube current, and 5 ms irradiation time conditions during 10 sec X-ray angiography in free air. This evaluated value was obtained by comparing each 30-degree measurement with the value of the 0° measurement. Furthermore, we determined the effect of backscattered radiation with and without 20 acrylic phantoms for the novel and previous sensors at 60, 80, 100, and 120 kV tube voltage, 250 mA tube current, and 200 ms irradiation time for the same exposure time. The evaluated values were obtained by averaging three measurements for the novel and previous sensors. The values of each sensor were evaluated by a ratio between the following: with acrylic phantoms (back scatterers) and without it. 

To evaluate the image quality for each dosimeter, we observed the novel and previous sensors and SDM placed on a chest phantom using a display monitor under 70 kV tube voltage, 61 mA tube current, and 6.9 ms irradiation time for the fluoroscopic image as X-ray conditions.

The above evaluation methods of 2.2 and 2.3 were performed according to our previous study [25].

## 3. Results

### 3.1. Basic Characteristics

The uniformity of the 4-channel novel sensors indicated a CV of 5.2% (Table 1). The reproducibility of each sensor indicated a CV of 0.3, 0.2, 0.2, 0.5% (ch.1, ch 2, ch 3, ch 4, respectively).

Figure 5 shows the energy (tube voltage) dependence, where the novel and previous sensor and SDM readings were normalized by the value of the ion chamber. Obviously, all three sensors perform well while the tube voltage is larger than 90 kV. Mean error (ME) of larger data points than 90 kV was +2.6, +0.1, −2.5% (respectively, novel, previously sensor and SDM). The previous sensor slightly indicated better than the novel sensor and SDM.

Figure 6 shows the correlations between the measured absorbed dose (novel, previous sensors and SDM) and the reference-absorbed dose (ion chamber measurement) as the dose-dependence in the logarithmic expression. All radiation sensors have excellent determination coefficients (R^2^ = 1.0000), although the SDM had slightly low sensitivity, below 1 mGy. 

Figure 7 shows the dose rate dependence as relative values, where the novel and previous sensor and SDM readings were normalized by the value of the ion chamber. Our radiation sensors indicated a remarkable dose rate dependence, although the SDM decreased by about 20% at low dose rates (0.3–2 mGy/s). 

### 3.2. Angular Dependence and Fluoroscopic Image

The angular dependences are shown for the horizontal (Figure 8) and vertical (Figure 9) axis for each dosimeter, normalized to the 0° measurement value. The novel sensor value of all angles was almost 100% sensitivity in both axes, except 270 degrees in the vertical axis. For with and without the 20 acrylic phantoms as back scatterers, the novel sensor indicated approximately 20% higher value than the previous sensor in the medical X-ray range (Figure 10).

Figure 11 shows the fluoroscopic images of the novel and previous sensors and SDM placed on chest phantom. The optical fiber cables of 2.5 m of our sensors and SDM were invisible in fluoroscopy. The novel sensor had a small size and shape, so it had less effect on X-ray images than the other radiation sensors. Moreover, we can confirm it to be in the irradiated field because of being slightly visible in the fluoroscopic images.

## 4. Discussion

Radiation-related tissue injuries of patients during FGI procedures have been reported in the literature [1,2]. It is essential for measuring radiation exposure in real-time to prevent deterministic effects, such as radiation skin injuries [27,28,29,30,31,32,33]. Therefore, we developed a novel real-time radiation system using a 4-channel radiation sensor, which allows the evaluation of maximum skin radiation exposure in patients undergoing FGI procedures.

This study was performed to clarify the effectiveness of our novel radiation sensors experimentally. Energy dependence of our novel and previous sensors had similar characteristics as the SDM for energy dependence, although the sensitivity tended to decrease marginally at 60 kV and 70 kV tube voltages. Consequently, our sensors need to use the energy calibration factor in the low-tube voltage field. We found excellent determination coefficients between all real-time dosimeter measurements and referenced ion chamber measurement from about 0.2–500 mGy (R^2^ = 1.0000). Both our radiation sensors demonstrated almost no dose rate dependence; thus, indicating they can handle various dose rates, and are highly useful (within 5% statistical difference). The sensitivity of the SDM decreased by about 20% in a low dose rate field (0.3–2 mGy/s), as compared to the ion chamber. Compared to the previous sensor, the angle dependence of the novel sensor improved dramatically, both in the horizontal and vertical axes (Figure 8 and Figure 9). Moreover, as shown in Figure 10, it can be used to accurately measure patient radiation exposure that includes back scattered radiation, such as FGI. The shape of the phosphor of the new radiation sensor was modified from a disk to a circular cone, and the size of that was slightly reduced from Φ1.4 × 1.2 mm to Φ1.0 × 1.2 mm (Figure 1 and Figure 2). Therefore, the novel sensor can emphasize constant sensitivity in all angles and be very useful for FGI, in which fluoroscopy is performed in various directions. 

In summary, the basic characteristics of the novel sensor were superior to other radiation sensors used for measuring radiation doses in patients. Furthermore, it had less influence on the X-ray images compared with the other radiation sensors because of modifying a small shape and size. We believe that the novel sensor proved to be critically useful in a real-time radiation system for FGI procedures.

Our work has some limitations. The present study only examined a phantom study. Consequently, we need to investigate not only the results of the phantom study but also the results of a clinical setting. Therefore, in the future, we should apply our novel sensor in a clinical setting in a situation for high doses and high dose rates, such as computed tomography.

## 5. Conclusions

The novel real-time radiation sensor we developed is expected to be useful for measuring the exposure of patients to radiation doses. In particular, the new sensor has constant sensitivity from all angles and does not clinically affect X-ray images by modifying shape and size. Therefore, the radiation sensor has the potential for patient skin dose exposure monitoring during FGI procedures.

## Figures and Tables

**Figure 1 sensors-20-02741-f001:**
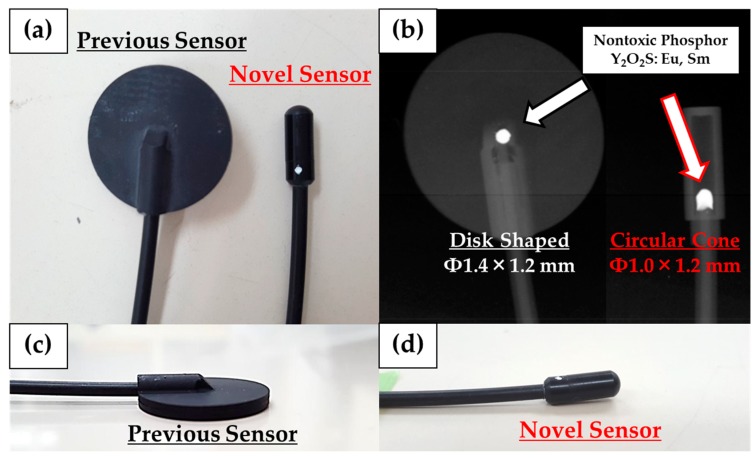
Novel and previous sensors: (**a**) top view, (**b**) X-ray image, (**c**) (**d**) lateral view. By changing the phosphor from a disk to cone shape, the novel dosimeter can provide better angular sensitivity.

**Figure 2 sensors-20-02741-f002:**
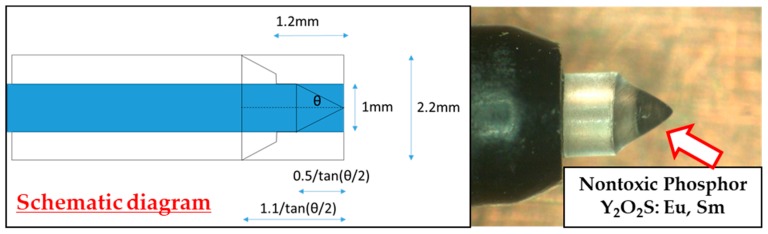
The internal structure of the novel sensor modified to a circular cone sensor.

**Figure 3 sensors-20-02741-f003:**
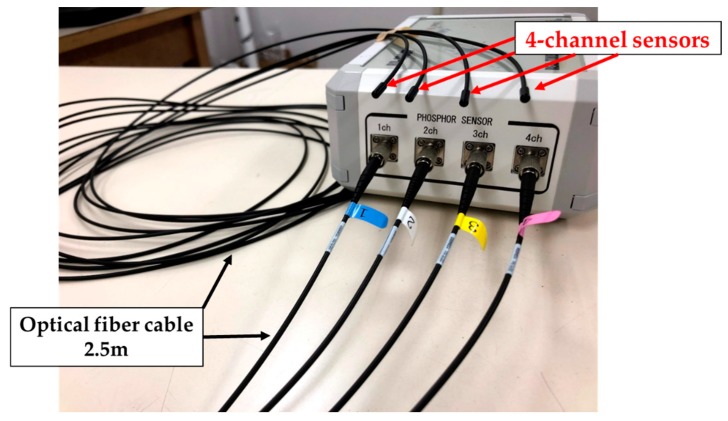
Apparent condition of novel real-time radiation system using 4-channel sensors.

**Figure 4 sensors-20-02741-f004:**
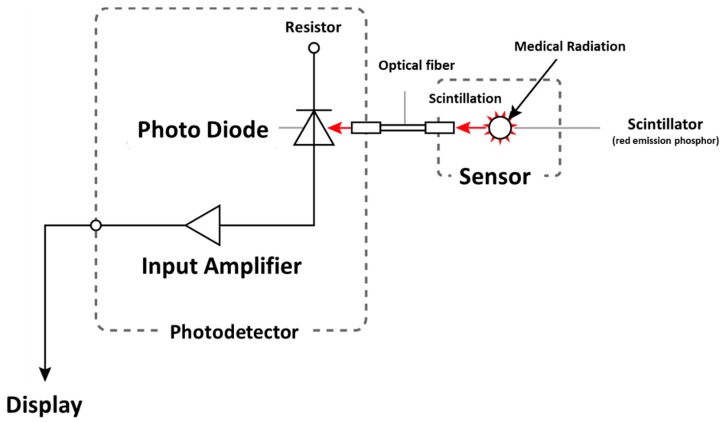
Operational principle of novel real-time radiation system.

**Figure 5 sensors-20-02741-f005:**
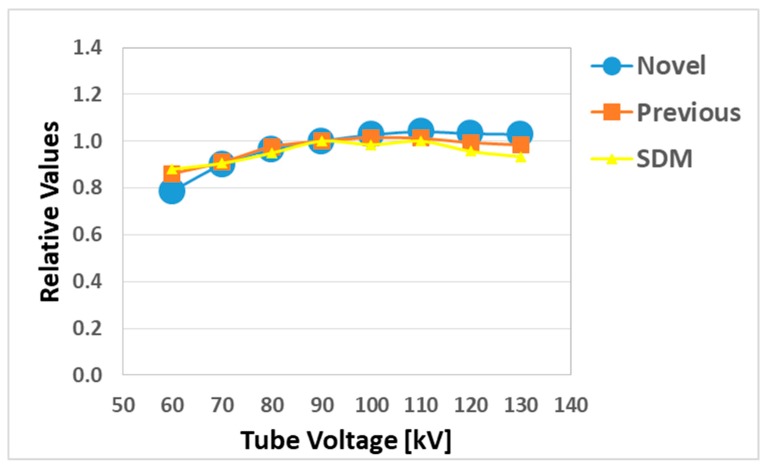
Energy (tube voltage) dependence compared with the novel, previous sensors and skin dose monitors (SDM).

**Figure 6 sensors-20-02741-f006:**
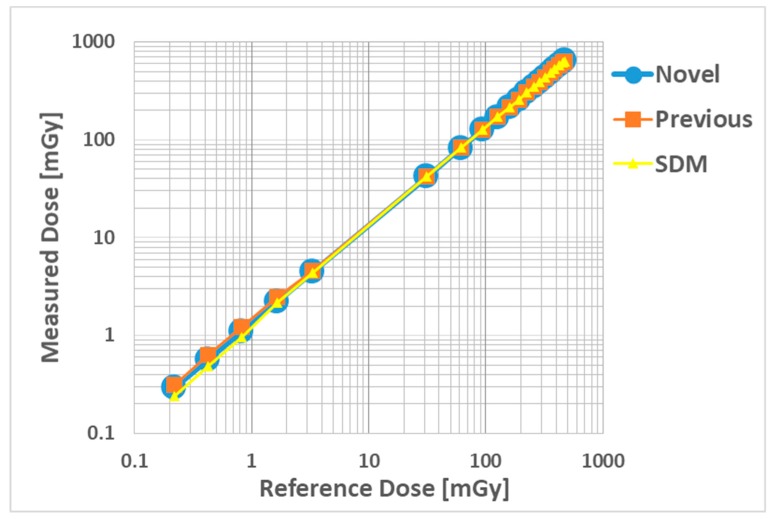
Dose dependence compared with the novel, previous sensors and SDM. Linear regression functions; novel dosimeter (R^2^ = 1.0000, y = 1.3501x - 0.2001), previous dosimeter (R^2^ = 1.0000, y = 1.3436x + 0.3608), SDM (R^2^ = 1.0000, y = 1.3471x + 0.0362). R^2^: determination coefficients.

**Figure 7 sensors-20-02741-f007:**
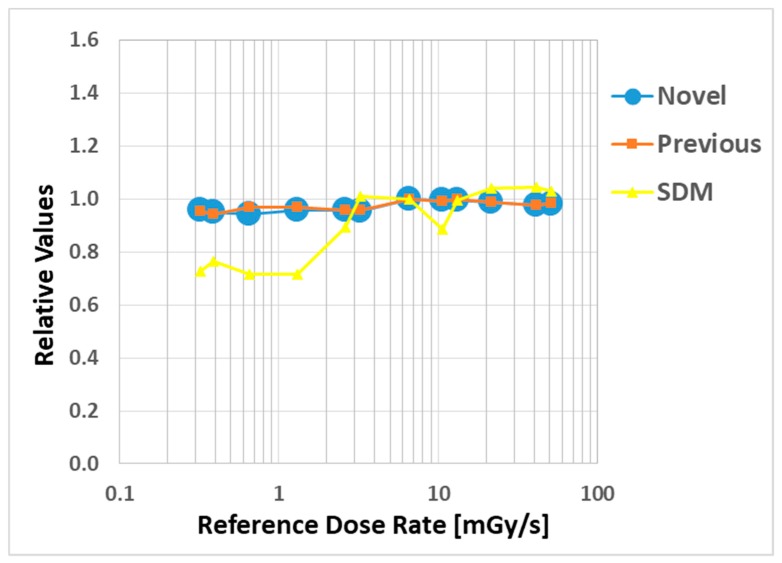
Dose rate dependence compared with the novel, previous sensors, and SDM.

**Figure 8 sensors-20-02741-f008:**
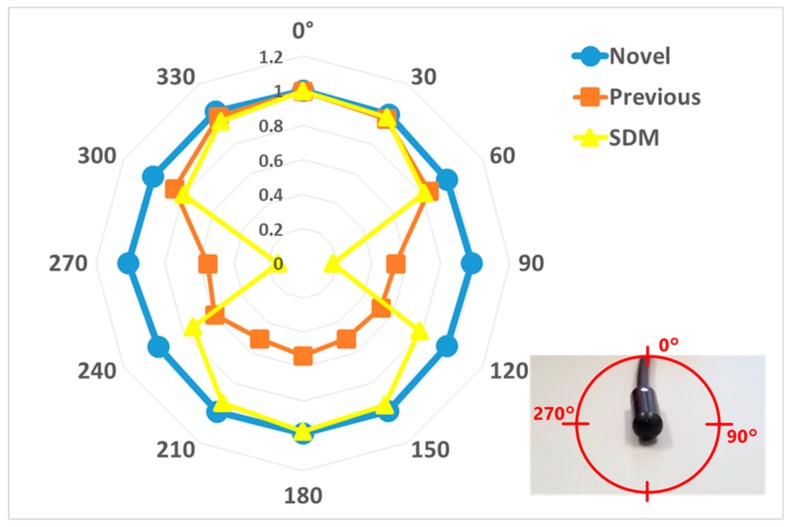
Angular dependence for horizontal axis compared with the novel, previous sensors and SDM.

**Figure 9 sensors-20-02741-f009:**
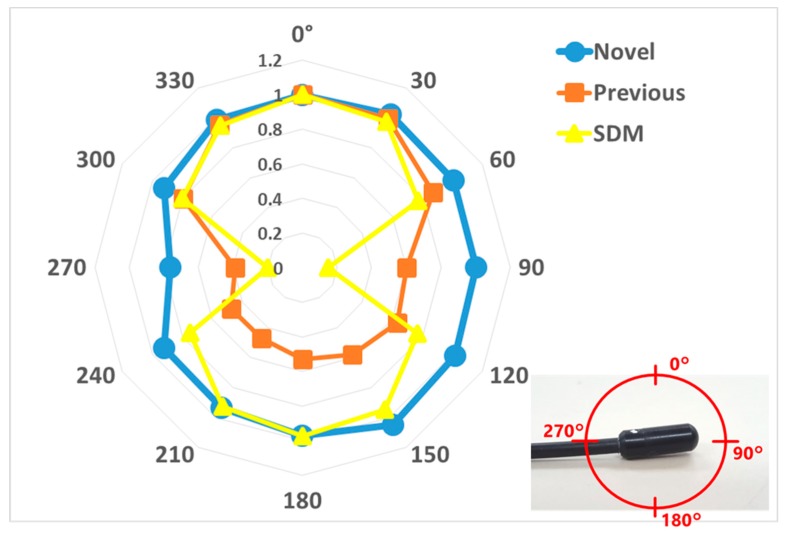
Angular dependence for vertical axis compared with the novel, previous sensors and SDM.

**Figure 10 sensors-20-02741-f010:**
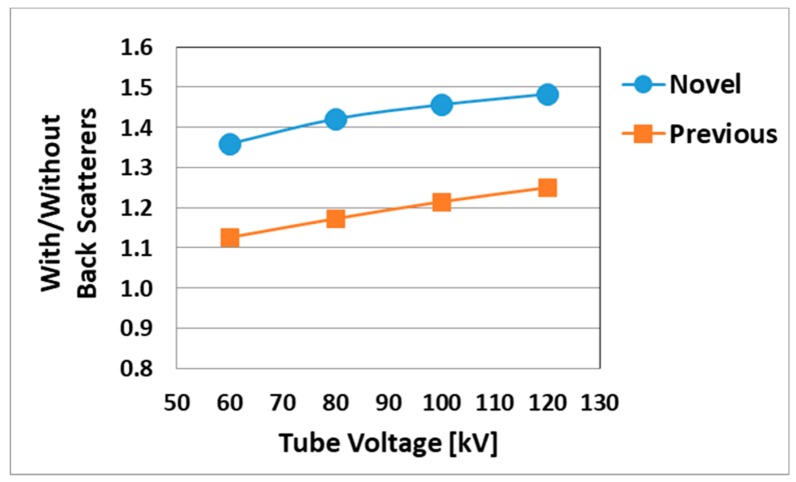
Comparison of with and without scatterers in the novel and previous sensors.

**Figure 11 sensors-20-02741-f011:**
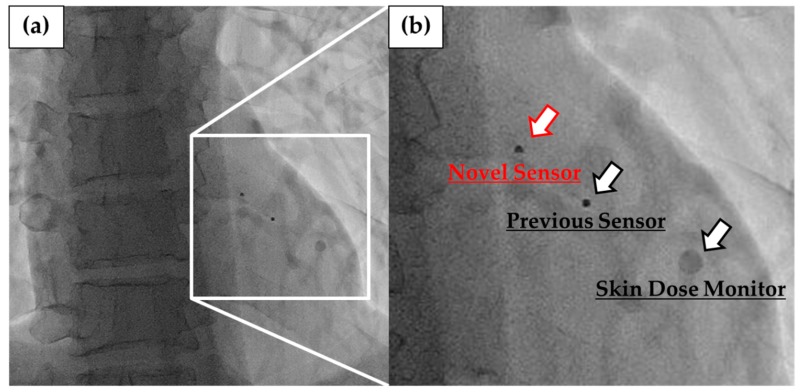
Fluoroscopic image: (**a**) 3 sensors placed on chest phantom, (**b**) magnified view.

**Table 1 sensors-20-02741-t001:** Uniformity of the 4-channel novel sensors.

Channel Number.	ch. 1	ch. 2	ch. 3	ch. 4	Ave.	SD	CV
Measurements	mGy				mGy		
**1**	3.88	4.18	4.27	3.87	4.05	0.205	0.051
**2**	3.84	4.16	4.25	3.82	4.02	0.220	0.055
**3**	3.85	4.17	4.25	3.83	4.03	0.216	0.054
**4**	3.86	4.17	4.26	3.84	4.03	0.214	0.053
**5**	3.86	4.18	4.26	3.84	4.04	0.216	0.054
**6**	3.87	4.18	4.27	3.85	4.04	0.214	0.053
**7**	3.87	4.18	4.26	3.85	4.04	0.211	0.052
**8**	3.87	4.17	4.26	3.86	4.04	0.205	0.051
**9**	3.87	4.18	4.27	3.87	4.05	0.208	0.051
**10**	3.87	4.18	4.27	3.87	4.05	0.208	0.051
**Ave.**	3.86	4.18	4.26	3.85	4.04	0.212	**0.052**
							**Uniformity**

**Ave.**: average, SD: standard deviation, CV: coefficient of variation, mGy: unit of radiation dose.

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
