# Peer review of "Development of Novel Real-Time Radiation Systems Using 4-Channel Sensors"

_sensors, 2020, doi:10.3390/s20092741_

Round 1
Reviewer 1 Report
In this manuscript, the authors proposed a novel real-time radiation system. The authors introduce the construction of the sensor system. Then the proposed sensor is compared with the authors-developed previous sensor and SMD for essential characteristics, including energy dependence, dose dependence, and dose rate independence to support the proposed sensor system is superior to the other two sensors. Besides, the authors also present and discuss the tests for angular dependence and the sensor size effect on the fluoroscopic image for all three sensors. Based on the sensor development experience, the studies for the new proposed sensor are functional; however, there are some fatal issues are funded in this manuscript and might not acceptable,
1. As the authors indicate in the title, the new proposed dosimeter system consists of four channels. In this paper, the authors discuss the sensor characteristics, the angular dependence, and the sensor on the fluoroscopy images are all based on a single sensor. It is not clear why the authors emphasize the dosimeter system consists of four novel sensors but without any discussion about how to implement a measurement with four sensors and the what is the performance as four sensors integrated.
2. Following the first question 1, how to implement the phosphor sensors into a sensing system and use for skin injury monitoring?
3. Figure 4 shows the uniformity and reproducibility of the 4-channel novel sensors; however, the authors fail to present the measurement conditions. How to take the test data is essential for evaluating the results of uniformity and reproducibility of the sensors? Meanwhile, according to general definitions, those two data should be obtained from different experiments operated under different conditions. We expect to see the details.
4. Figure 5 is about tube voltage dependence; obviously, all three sensors perform well while the tube voltage is large than 90 V.
Without having the detailed data, but intuitively comparing the data points over 90 V, the previous sensor developed by the authors is better than sensor proposed in this manuscript and SDM because of the departures from 1.0 is the smallest among all the three sensors.
5. Line 138-140, the authors should emphasize the fitted curves are in the logarithm scale. The authors remark that R2= 1.00 of the fitted curves for all sensors; however, the most critical parameter is not the determination of coefficient but the fitting functions. The authors should provide the regression functions of three methods for comparison.
Author Response
"Please see the attachment."

Reviewer 2 Report
The authors present a novel four-channel real-time radiation system for monitoring of medical radiation exposure during fluoroscopically guided intervention (FGI). The proposed system is a significant improvement in the present version. The authors evaluated the novel sensor in comparison with the previous version and with a skin dose monitor. A 6ml thimble ion chamber was used for obtaining a reference value.
The topic is within the scope of the journal. The paper is well prepared, fully understandable and well structured. I recommend accepting the paper.
Minor Comments:
- Please, try a little bit different marks in figures 5, 6 and 7 to improve the readability of the figures.
Author Response
"Please see the attachment."

Reviewer 3 Report
The authors developed and characterized their second multi-channel dosimeter target for dose monitoring during FGI process. By comparing the results with an ion chamber, the new dosimeter shows a much better angular sensitivity. The methods, results and conclusions are complete and coherent. The background and language are marginal. Here I have some minor comments:
Page 1 line 18. Change to… we developed a novel 4-channel sensor with modified shape and size than …
Page 1, line 41, before introducing the developed dosimeter, please briefly provide the current dosimeters or technologies to monitor the dose in real time during FGI.
Page 2, line 47, somewhere in this paragraph, please add the operational principle of this type of dosimeter.
Page 2, line 55, when writing a number with a unit, there should be a space between the value and the unit, like 2.5 m. Please change all the others in the manuscript.
Page 3, line 83. The subtitle of Basic Characteristics is too broad, using anther more meaningful name such as Dosimeter Response characteristics.
Page 3, line 98, what is the energy inside the bracket? Is it the mean energies?
Page 3, line 97-101. Change to: the energy dependence experiment of ….. The evaluated values were obtained by averaging three measurements for each dosimeter. The ion chamber results were used as a reference and all the values were normalized to its 90 kV dose.
Page 3, line 112. Change to: The X-ray unit used for angular dependent measurement and …
Page 4, line 119, … and previous sensors at 60, 80, …
Page 4, line 121, the last sentence is confusing.
Page 4, Figure 4, this is an table, not a figure. Change 1ch, 2ch ect. To ch1, ch2,…
Page 4, figure 5. The legends are Novel, Previous and SDM, but the author mentioned in section .2, the previous sensor is the SDM, and the ion chamber results are also used. So what is the different between the previous sensor and the SDM, and where is the ion chamber results. Please also clarify the legends in the following figures: figure 6-figure9.
Page 5, figure 6. When you talking about the determination coefficient, you mean the measured data has a perfect linear relation with the reference dose. If this is the case, you may want to use 4 decimal point value to result the R^2. Because by looking at the curve, at least the SDM results is not perfectly linear.
Page 5, line 146, change to: …by about 20% at low dose rates (0.3-2 mGy/s).
Page 7, line 172, change to: The optical fiber cables of 2.5 m …
Page 8, line 199. Maybe you want to delete the last sentence. Conduct the experiment using the previous procedure dose not guarantee a reliable study.
Page 9, conclusions: … the novel real-time radiation sensor we developed are expected …..Therefore, the radiation sensor has the potential for patient skin dose exposure monitoring during FGI procedures.
For all the title of the figures and tables, it is better to add a short summary of the message in addition to the simple description. Like in figure 1. You can add: By changing the phosphor from a disk to cone shape, the new dosimeter can provide better angular sensitivity.
Author Response
"Please see the attachment."

Round 2
Reviewer 1 Report
- I am not satisfied with the replies about the sensor uniformity and reproducibility since there are no clear definitions given by the authors.
If the uniformity is about the performance deviation among the sensors, then the measurement must execute as all the four sensors at the corresponding positions and subject to the same testing condition; unfortunately, the authors fail to give the detailed measurement procedures and layout. I am not able to know whether the authors placed four sensors at fixed positions for each measurement, or the authors take data sequentially. Yet, according to the reply from authors, the measurement data is very likely obtained sequentially, which is against the general understanding of "uniformity."
Regarding the reproducibility, if the authors consider the four sensors as four individuals to measure the radiation dosage emitted at the same tube voltage at a different time. The measurements must execute sequentially and relocate the sensor for every single measuring. According to the authors' reply, the authors evaluate the sensors very likely in this way. But, whether the authors have relocated the sensor is unknown. Nevertheless, the authors cannot use the same data set for evaluating the uniformity and reproducibility that is against the definitions mentioned above, if I am right. Meanwhile, the authors calculate the SD for reproducibility calculation by taking an average of the single SD of 10 times measurements for each sensor; please confirm the correctness of the model. The SD of a population should be taking a square-sum of four SD of the four sensors and followed by dividing with the degree of freedom and then taking square-root.
The question is essential for evaluating the characteristics of the newly developed sensor and the value of the research; please enhance the discussion properly. - Minor question, regarding the linear regression functions, does the offset mean anything for the sensors?
Author Response
"Please see the attachment."

Round 3
Reviewer 1 Report
I appreciate the authors take action on changing the content. No more technical concern, I will be more than happy to suggest the manuscript to be published.